# DeepProbLog:
# Neural Probabilistic Logic Programming

**Robin Manhaeve**
KU Leuven
robin.manhaeve@cs.kuleuven.be

**Sebastijan Dumančić**
KU Leuven
sebastijan.dumancic@cs.kuleuven.be

**Angelika Kimmig**
Cardiff University
KimmigA@cardiff.ac.uk

**Thomas Demeester**[*]
Ghent University - imec
thomas.demeester@ugent.be

**Luc De Raedt**[*]
KU Leuven
luc.deraedt@cs.kuleuven.be

## Abstract

We introduce DeepProbLog, a probabilistic logic programming language that incorporates deep learning by means of neural predicates. We show how existing inference and learning techniques can be adapted for the new language. Our experiments demonstrate that DeepProbLog supports (i) both symbolic and sub-symbolic representations and inference, (ii) program induction, (iii) probabilistic (logic) programming, and (iv) (deep) learning from examples. To the best of our knowledge, this work is the first to propose a framework where general-purpose neural networks and expressive probabilistic-logical modeling and reasoning are integrated in a way that exploits the full expressiveness and strengths of both worlds and can be trained end-to-end based on examples.

## 1   Introduction

The integration of low-level perception with high-level reasoning is one of the oldest, and yet most current open challenges in the field of artificial intelligence. Today, low-level perception is typically handled by neural networks and deep learning, whereas high-level reasoning is typically addressed using logical and probabilistic representations and inference. While it is clear that there have been breakthroughs in deep learning, there has also been a lot of progress in the area of high-level reasoning. Indeed, today there exist approaches that tightly integrate logical and probabilistic reasoning with statistical learning; cf. the areas of statistical relational artificial intelligence [De Raedt et al., 2016, Getoor and Taskar, 2007] and probabilistic logic programming [De Raedt and Kimmig, 2015]. Recently, a number of researchers have revisited and modernized older ideas originating from the field of neural-symbolic integration [Garcez et al., 2012], searching for ways to combine the best of both worlds [Bošnjak et al., 2017, Rocktäschel and Riedel, 2017, Cohen et al., 2018, Santoro et al., 2017], for example, by designing neural architectures representing differentiable counterparts of symbolic operations in classical reasoning tools. Yet, joining the full flexibility of high-level probabilistic reasoning with the representational power of deep neural networks is still an open problem. This paper tackles this challenge from a different perspective. Instead of integrating reasoning capabilities into a complex neural network architecture, we proceed the other way round. We start

---

[*] joint last authors

from an existing probabilistic logic programming language, ProbLog [De Raedt et al., 2007], and extend it with the capability to process neural predicates. The idea is simple: in a probabilistic logic, atomic expressions of the form $q(t_1, ..., t_n)$ (aka tuples in a relational database) have a probability $p$. Consequently, the output of neural network components can be encapsulated in the form of "neural" predicates as long as the output of the neural network on an atomic expression can be interpreted as a probability. This simple idea is appealing as it allows us to retain all the essential components of the ProbLog language: the semantics, the inference mechanism, as well as the implementation. The main challenge is in training the model based on examples. The input data consists of feature vectors at the input of the neural network components (e.g., images) together with other probabilistic facts and clauses in the logic program, whereas targets are only given at the output side of the probabilistic reasoner. However, the algebraic extension of ProbLog (based on semirings) [Kimmig et al., 2011] already supports automatic differentiation. As a result, we can back-propagate the gradient from the loss at the output through the neural predicates into the neural networks, which allows training the whole model through gradient-descent based optimization. We call the new language *DeepProbLog*.

Before going into further detail, the following example illustrates the possibilities of this approach (also see Section 6). Consider the predicate $\texttt{addition}(\texttt{X}, \texttt{Y}, \texttt{Z})$, where $\texttt{X}$ and $\texttt{Y}$ are images of digits and $\texttt{Z}$ is the natural number corresponding to the sum of these digits. After training, DeepProbLog allows us to make a probabilistic estimate on the validity of, e.g., the example $\texttt{addition}(\text{🖩}, \text{🖩}, 8)$. While such a predicate can be directly learned by a standard neural classifier, such a method would have a hard time taking into account background knowledge such as the definition of the addition of two *natural* numbers. In DeepProbLog such knowledge can easily be encoded in rules such as $\texttt{addition}(\texttt{I}_\texttt{X}, \texttt{I}_\texttt{Y}, \texttt{N}_\texttt{Z}) := \texttt{digit}(\texttt{I}_\texttt{X}, \texttt{N}_\texttt{X}), \texttt{digit}(\texttt{I}_\texttt{Y}, \texttt{N}_\texttt{Y}), \texttt{N}_\texttt{Z} \text{ is } \texttt{N}_\texttt{X} + \texttt{N}_\texttt{Y}$ (with is the standard operator of logic programming to evaluate arithmetic expressions). All that needs to be learned in this case is the neural predicate $digit$ which maps an image of a digit $I_D$ to the corresponding natural number $N_D$. The learned network can then be reused for arbitrary tasks involving digits. Our experiments show that this leads not only to new capabilities but also to significant performance improvements. An important advantage of this approach compared to standard image classification settings is that it can be extended to multi-digit numbers without additional training. We note that the single digit classifier (i.e., the neural predicate) is not explicitly trained by itself: its output can be considered a latent representation, as we only use training data with pairwise sums of digits.

To summarize, we introduce DeepProbLog which has a unique set of features: (i) it is a programming language that supports neural networks and machine learning, and it has a well-defined semantics (as an extension of Prolog, it is Turing equivalent); (ii) it integrates logical reasoning with neural networks; so both symbolic and subsymbolic representations and inference; (iii) it integrates probabilistic modeling, programming and reasoning with neural networks (as DeepProbLog extends the probabilistic programming language ProbLog, which can be regarded as a very expressive directed graphical modeling language [De Raedt et al., 2016]); (iv) it can be used to learn a wide range of probabilistic logical neural models from examples, including inductive programming. The code is available at $\texttt{https://bitbucket.org/problog/deepproblog}$.

## 2 Logic programming concepts

We briefly summarize basic logic programming concepts. Atoms are expressions of the form $q(t_1, ..., t_n)$ where $q$ is a predicate (of arity $n$) and the $t_i$ are terms. A literal is an atom or the negation $\neg q(t_1, ..., t_n)$ of an atom. A term $t$ is either a constant $c$, a variable $V$, or a structured term of the form $f(u_1, ..., u_k)$ where $f$ is a functor. We will follow the Prolog convention and let constants start with a lower case character and variables with an upper case. A substitution $\theta = \{V_1 = t_1, ..., V_n = t_n\}$ is an assignment of terms $t_i$ to variables $V_i$. When applying a substitution $\theta$ to an expression $e$ we simultaneously replace all occurrences of $V_i$ by $t_i$ and denote the resulting expression as $e\theta$. Expressions that do not contain any variables are called ground. A rule is an expression of the form $h := b_1, ..., b_n$ where $h$ is an atom and the $b_i$ are literals. The meaning of such a rule is that $h$ holds whenever the conjunction of the $b_i$ holds. Thus $:-$ represents logical implication ($\leftarrow$), and the comma (,) represents conjunction ($\wedge$). Rules with an empty body $n = 0$ are called facts.

# 3 Introducing DeepProbLog

We now recall the basics of probabilistic logic programming using ProbLog, illustrate it using the well-known burglary alarm example, and then introduce our new language DeepProbLog.

A ProbLog program consists of (i) a set of ground probabilistic facts $\mathcal{F}$ of the form $p :: f$ where $p$ is a probability and $f$ a ground atom and (ii) a set of rules $\mathcal{R}$. For instance, the following ProbLog program models a variant of the well-known alarm Bayesian network:

$$0.1 :: \texttt{burglary}. \quad 0.5 :: \texttt{hears\_alarm(mary)}.$$
$$0.2 :: \texttt{earthquake}. \quad 0.4 :: \texttt{hears\_alarm(john)}.$$

$$\texttt{alarm :}- \texttt{earthquake}.$$
$$\texttt{alarm :}- \texttt{burglary}. \quad (1)$$
$$\texttt{calls(X) :}- \texttt{alarm}, \texttt{hears\_alarm(X)}.$$

Each probabilistic fact corresponds to an *independent Boolean random variable* that is true with probability $p$ and false with probability $1 - p$. Every subset $F \subseteq \mathcal{F}$ defines a possible world $w_F = F \cup \{f\theta | \mathcal{R} \cup F \models f\theta$ and $f\theta$ is ground$\}$. So $w_F$ contains $F$ and all ground atoms that are logically entailed by $F$ and the set of rules $\mathcal{R}$, e.g.,

$$w_{\{\texttt{burglary,hears\_alarm(mary)}\}} = \{\texttt{burglary}, \texttt{hears\_alarm(mary)}\} \cup \{\texttt{alarm}, \texttt{calls(mary)}\}$$

The probability $P(w_F)$ of such a possible world $w_F$ is given by the product of the probabilities of the truth values of the probabilistic facts, $P(w_F) = \prod_{f_i \in F} p_i \prod_{f_i \in \mathcal{F} \setminus F} (1 - p_i)$. For instance,

$$P(w_{\{\texttt{burglary,hears\_alarm(mary)}\}}) = 0.1 \times 0.5 \times (1 - 0.2) \times (1 - 0.4) = 0.024$$

The probability of a ground fact $q$, also called *success probability of $q$*, is then defined as the sum of the probabilities of all worlds containing $q$, i.e., $P(q) = \sum_{F \subseteq \mathcal{F} : q \in w_F} P(w_F)$.

One convenient extension that is nothing else than syntactic sugar are the annotated disjunctions. An annotated disjunction (AD) is an expression of the form $p_1 :: h_1; ...; p_n :: h_n :- b_1, ..., b_m$. where the $p_i$ are probabilities so that $\sum p_i = 1$, and $h_i$ and $b_j$ are atoms. The meaning of an AD is that whenever all $b_i$ hold, $h_j$ will be true with probability $p_j$, with all other $h_i$ false (unless other parts of the program make them true). This is convenient to model choices between different categorical variables, e.g. different severities of the earthquake:

$$0.4 :: \texttt{earthquake(none)}; 0.4 :: \texttt{earthquake(mild)}; 0.2 :: \texttt{earthquake(severe)}.$$

ProbLog programs with annotated disjunctions can be transformed into equivalent ProbLog programs without annotated disjunctions (cf. De Raedt and Kimmig [2015]).

A **DeepProbLog** program is a ProbLog program that is extended with (iii) a set of ground neural ADs (nADs) of the form

$$nn(m_q, \vec{t}, \vec{u}) :: q(\vec{t}, u_1); ...; q(\vec{t}, u_n) :- b_1, ..., b_m$$

where the $b_i$ are atoms, $\vec{t} = t_1, \ldots, t_k$ is a vector of ground terms representing the inputs of the neural network for predicate $q$, $u_1$ to $u_n$ are the possible output values of the neural network. We use the notation $nn$ for 'neural network' to indicate that this is a nAD. $m_q$ is the identifier of a neural network model that specifies a probability distribution over its output values $\vec{u}$, given input $\vec{t}$. That is, from the perspective of the probabilistic logic program, an nAD realizes a regular AD $p_1 :: q(\vec{t}, u_1); ...; p_n :: q(\vec{t}, u_n) :- b_1, ..., b_m$, and DeepProbLog thus directly inherits its semantics, and to a large extent also its inference, from ProbLog. For instance, in the MNIST addition example, we would specify the nAD

$$nn(m_{\text{digit}}, \text{🄵}, [0, \ldots, 9]) :: \texttt{digit}(\text{🄵}, 0); \ldots; \texttt{digit}(\text{🄵}, 9).$$

where $m_{\text{digit}}$ is a network that probabilistically classifies MNIST digits. The neural network could take any shape, e.g., a convolutional network for image encoding, a recurrent network for sequence encoding, etc. However, its output layer, which feeds the corresponding neural predicate, needs to be normalized. In neural networks for multiclass classification, this is typically done by applying a softmax layer to real-valued output scores, a choice we also adopt in our experiments.

## 4 DeepProbLog Inference

This section explains how a DeepProbLog model is used for a given query at prediction time.

**ProbLog Inference**  As inference in DeepProbLog closely follows that in ProbLog, we now summarize ProbLog inference using the burglary example explained before. For full details, we refer to Fierens et al. [2015]. The program describing the example is explained in Section 3, Equation (1). We can query the program for the probabilities of given query atoms, say, the single query `calls(mary)`. ProbLog inference proceeds in four steps. (i) The first step grounds the logic program with respect to the query, that is, it generates all ground instances of clauses in the program the query depends on. The grounded program for query `calls(mary)` is shown in Figure 1a. (ii) The second step rewrites the ground logic program into a formula in propositional logic that defines the truth value of the query in terms of the truth values of probabilistic facts. In our example, the resulting formula is `calls(mary)` ↔ `hears_alarm(mary)` ∧ (`burglary` ∨ `earthquake`). (iii) The third step compiles the logic formula into a Sentential Decision Diagram (SDD, Darwiche [2011]), a form that allows for efficient evaluation of the query, using knowledge compilation technology [Darwiche and Marquis, 2002]. The SDD for our example is shown in Figure 1b, where rounded grey rectangles depict variables corresponding to probabilistic facts, and the rounded red rectangle denotes the query atom defined by the formula. The white rectangles correspond to logical operators applied to their children. (iv) The fourth and final step evaluates the SDD bottom-up to calculate the success probability of the given query, starting with the probability labels of the leaves as given by the program and performing addition in every or-node and multiplication in every and-node. The intermediate results are shown next to the nodes in Figure 1b, ignoring the blue numbers for now.

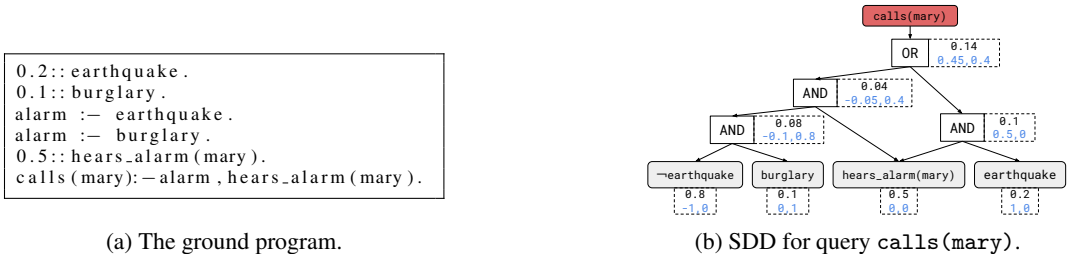

```
0.2:: earthquake .
0.1:: burglary .
alarm :−  earthquake .
alarm :−  burglary .
0.5:: hears_alarm ( mary ).
calls ( mary ):−alarm , hears_alarm ( mary ).
```

(a) The ground program.           (b) SDD for query `calls(mary)`.

Figure 1: Inference in ProbLog.

**DeepProbLog Inference**  Inference in DeepProbLog works exactly as described above, except that a forward pass on the neural network components is performed every time we encounter a neural predicate during grounding. When this occurs, the required inputs (e.g., images) are fed into the neural network, after which the resulting scores of their softmax output layer are used as the probabilities of the ground AD.

## 5 Learning in DeepProbLog

We now introduce our approach to jointly train the parameters of probabilistic facts and neural networks in DeepProbLog programs. We use the *learning from entailment* setting [De Raedt et al., 2016] , that is, given a DeepProbLog program with parameters $\mathcal{X}$, a set $\mathcal{Q}$ of pairs $(q, p)$ with $q$ a query and $p$ its desired success probability, and a loss function $L$, compute:

$$\arg\min_{\vec{x}} \frac{1}{|\mathcal{Q}|} \sum_{(q,p) \in \mathcal{Q}} L(P_{\mathcal{X}=\vec{x}}(q), p)$$

In contrast to the earlier approach for ProbLog parameter learning in this setting by Gutmann et al. [2008], we use gradient descent rather than EM, as this allows for seamless integration with neural network training. An overview of the approach is shown in Figure 2a. Given a DeepProbLog program, its neural network models, and a query used as training example, we first ground the program with respect to the query, getting the current parameters of nADs from the external models, then use the ProbLog machinery to compute the loss and its gradient, and finally use these to update the parameters in the neural networks and the probabilistic program.

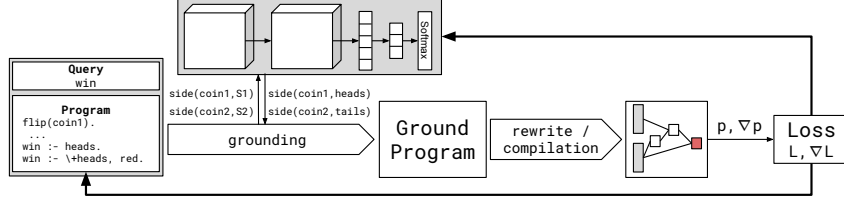

(a) The learning pipeline.

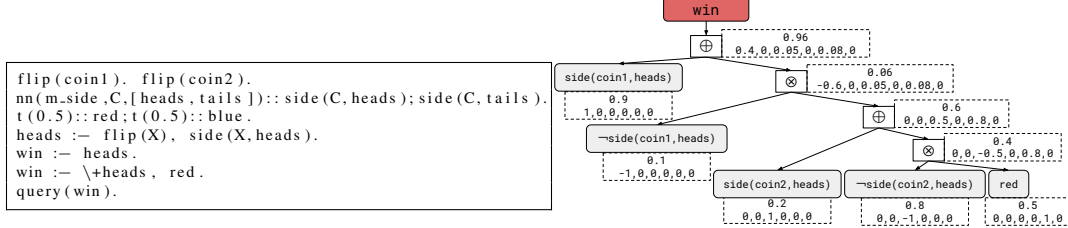

(b) The DeepProbLog program.  (c) SDD for query `win`.

Figure 2: Parameter learning in DeepProbLog.

More specifically, to compute the gradient with respect to the probabilistic logic program part, we rely on Algebraic ProbLog (aProbLog, [Kimmig et al., 2011]), a generalization of the ProbLog language and inference to arbitrary commutative semirings, including the gradient semiring [Eisner, 2002]. In the following, we provide the necessary background on aProbLog, discuss how to use it to compute gradients with respect to ProbLog parameters and extend the approach to DeepProbLog.

**aProbLog and the gradient semiring**  ProbLog annotates each probabilistic fact $f$ with the probability that $f$ is true, which implicitly also defines the probability that $f$ is false, and thus its negation $\neg f$ is true. It then uses the probability semiring with regular addition and multiplication as operators to compute the probability of a query on the SDD constructed for this query, cf. Figure 1b. This idea is generalized in aProbLog to compute such values based on arbitrary commutative semirings. Instead of probability labels on facts, aProbLog uses a labeling function that explicitly associates values from the chosen semiring with both facts and their negations, and combines these using semiring addition $\oplus$ and multiplication $\otimes$ on the SDD. We use the gradient semiring, whose elements are tuples $(p, \frac{\partial p}{\partial x})$, where $p$ is a probability (as in ProbLog), and $\frac{\partial p}{\partial x}$ is the partial derivative of that probability with respect to a parameter $x$, that is, the probability $p_i$ of a probabilistic fact with learnable probability, written as $t(p_i) :: f_i$. This is easily extended to a vector of parameters $\vec{x} = [x_1, \ldots, x_N]^T$, the concatenation of all $N$ parameters in the ground program. Semiring addition $\oplus$, multiplication $\otimes$ and the neutral elements with respect to these operations are defined as follows:

$$(a_1, \vec{a_2}) \oplus (b_1, \vec{b_2}) = (a_1 + b_1, \vec{a_2} + \vec{b_2}) \quad (2) \qquad e^{\oplus} = (0, \vec{0}) \quad (4)$$

$$(a_1, \vec{a_2}) \otimes (b_1, \vec{b_2}) = (a_1 b_1, b_1 \vec{a_2} + a_1 \vec{b_2}) \quad (3) \qquad e^{\otimes} = (1, \vec{0}) \quad (5)$$

Note that the first element of the tuple mimics ProbLog's probability computation, whereas the second simply computes gradients of these probabilities using derivative rules.

**Gradient descent for ProbLog**  To use the gradient semiring for gradient descent parameter learning in ProbLog, we first transform the ProbLog program into an aProbLog program by extending the label of each probabilistic fact $p :: f$ to include the probability $p$ as well as the gradient vector of $p$ with respect to the probabilities of all probabilistic facts in the program, i.e.,

$$L(f) = (p, \vec{0}) \qquad \text{for } p :: f \text{ with fixed } p \quad (6)$$
$$L(f_i) = (p_i, \mathbf{e}_i) \qquad \text{for } t(p_i) :: f_i \text{ with learnable } p_i \quad (7)$$
$$L(\neg f) = (1 - p, -\nabla p) \qquad \text{with } L(f) = (p, \nabla p) \quad (8)$$

where the vector $\mathbf{e}_i$ has a 1 in the $i$th position and 0 in all others. For fixed probabilities, the gradient does not depend on any parameters and thus is 0. For the other cases, we use the semiring labels as introduced above. For instance, assume we want to learn the probabilities of earthquake

and `burglary` in the example of Figure 1, while keeping those of the other facts fixed. Then, in Figure 1b, the nodes in the SDD now also contain the gradient (below, in blue). The result shows that the partial derivative of the proof query is $0.45$ and $0.4$ w.r.t. the earthquake and burglary parameters respectively. To ensure that ADs are always well defined, i.e., the probabilities of the facts in the same AD sum to one, we re-normalize these after every gradient descent update.

**Gradient descent for DeepProbLog**    In contrast to probabilistic facts and ADs, whose parameters are updated based on the gradients computed by aProbLog, the probabilities of the neural predicates are a function of the neural network parameters. The neural predicates serve as an interface between the logic and the neural side, with both sides treating the other as a black box. The logic side can calculate the gradient of the loss w.r.t. the output of the neural network, but is unaware of the internal parameters. However, the gradient w.r.t. the output is sufficient to start backpropagation, which calculates the gradient for the internal parameters. Then, standard gradient-based optimizers (e.g. SGD, Adam, ...) are used to update the parameters of the network. During gradient computation with aProbLog, the probabilities of neural ADs are kept constant. Furthermore, updates on neural ADs come from the neural network part of the model, where the use of a softmax output layer ensures they always represent a normalized distribution, hence not requiring the additional normalization as for non-neural ADs. The labeling function for facts in nADs is

$$L(f_j) = (m_q(\vec{t})_j, \mathbf{e}_j) \qquad \text{for } nn(m_q, \vec{t}, \vec{u}) :: f_i; \ldots; f_k \text{ a nAD} \qquad (9)$$

**Example**    We will demonstrate the learning pipeline (shown in Figure 2a) using the following game. We have two coins and an urn containing red and blue balls. We flip both coins and take a ball out of the urn. We win if the ball is red, or at least one coin comes up heads. However, we need to learn to recognize heads and tails using a neural network, while only observing examples of wins or losses instead of explicitly learning from coin examples labeled with the correct side. We also learn the distribution of the red and blue balls in the urn. We show the program in Figure 2b. There are 6 parameters in this program: the first four originate from the neural predicates (heads and tails for the first and second coin). The last two are the logic parameters that model the chance of picking out a red or blue ball. During grounding, the neural network classifies `coin1` and `coin2`. According to the neural network, the first coin is most likely heads ($p = 0.9$), and the second one most likely tails ($p = 0.2$). Figure 2c shows the corresponding SDD with the AND/OR nodes replaced with the respective semiring operations. The top number is the probability, and the numbers below are the gradient. On the top node, we can see that we have a $0.96$ probability of winning, but also that the gradient of this probability is $0.4$ and $0.05$ w.r.t the probability of being heads for the first and second coin respectively, and $0.08$ for the chance of the ball being red.

## 6    Experimental Evaluation

We perform three sets of experiments to demonstrate that DeepProbLog supports (i) symbolic and subsymbolic reasoning and learning, that is, both logical reasoning and deep learning; (ii) program induction; and (iii) both probabilistic logic programming and deep learning.

We provide implementation details at the end of this section and list all programs in Appendix A.

**Logical reasoning and deep learning**    To show that DeepProbLog supports both logical reasoning and deep learning, we extend the classic learning task on the MNIST dataset (Lecun et al. [1998]) to two more complex problems that require reasoning:

**T1:** `addition(`3̸`, `5̸`, 8)`: Instead of using labeled single digits, we train on pairs of images, labeled with the sum of the individual labels. The DeepProbLog program consists of the clause `addition(X,Y,Z) :- digit(X,X2), digit(Y,Y2), Z is X2+Y2.` and a neural AD for the `digit/2` predicate (this is shorthand notation for *of arity 2*), which classifies an MNIST image. We compare to a CNN baseline classifying the concatenation of the two images into the 19 possible sums.

**T2:** `addition([`3̸`, `8̸`], [`2̸`, `5̸`], 63)`: the input consists of two lists of images, each element being a digit. This task demonstrates that DeepProbLog generalizes well beyond training data. Learning the new predicate requires only a small change in the logic program. We train the model on single digit numbers, and evaluate on three digit numbers.

The learning curves of both models on **T1** (Figure 3a) show the benefit of combined symbolic and subsymbolic reasoning: the DeepProbLog model uses the encoded knowledge to reach a higher F1 score than the CNN, and does so after a few thousand iterations, while the CNN converges much slower. We also tested an alternative for the neural network baseline. It evaluates convolutional layers with shared parameters on each image separately, instead of a single set of convolutional layers on the concatenation of both images. It converges quicker and achieves a higher final accuracy than the other baseline, but is still slower and less accurate than the DeepProbLog model. Figure 3b shows the learning curve for **T2**. DeepProbLog achieves a somewhat lower accuracy compared to the single digit problem due to the compounding effect of the error, but the model generalizes well. The CNN does not generalize to this variable-length problem setting.

**Program Induction** The second set of problems demonstrates that DeepProbLog can perform program induction. We follow the program sketch setting of differentiable Forth [Bošnjak et al., 2017], where holes in given programs need to be filled by neural networks trained on input-output examples for the entire program. As in their work, we consider three tasks: addition, sorting [Reed and de Freitas, 2016] and word algebra problems (WAPs) [Roy and Roth, 2015].

**T3:** `forth_addition/4`: where the input consists of two numbers and a carry, with the output being the sum of the numbers and the new carry. The program specifies the basic addition algorithm in which we go from right to left over all digits, calculating the sum of two digits and taking the carry over to the next pair. The hole in this program corresponds to calculating the resulting digit (`result/4`) and carry (`carry/4`), given two digits and the previous carry.

**T4:** `sort/2`: The input consists of a list of numbers, and the output is the sorted list. The program implements bubble sort, but leaves open what to do on each step in a bubble (i.e. whether to swap or not, `swap/2`).

**T5:** `wap/2`: The input to the WAPs consists of a natural language sentence describing a simple mathematical problem, and the output is the solution to this question. These WAPs always contain three numbers and are solved by chaining 4 steps: permuting the three numbers (`permute/2`), applying an operation on the first two numbers (addition, subtraction or product `operation_1/2`), potentially swapping the intermediate result and the last digit (`swap/2`), and performing a last operation (`operation_2/2`). The hole in the program is in deciding which of the alternatives should happen on each step.

DeepProbLog achieves 100% on the Forth addition (**T3**) and sorting (**T4**) problems (Table 1a). The sorting problem yields a more interesting comparison: differentiable Forth achieves a 100% accuracy with a training length of 2 and 3, but performs poorly on a training length of 4; DeepProbLog generalizes well to larger lengths. As shown in Table 1b, DeepProbLog runs faster and scales better with increasing training length, while differentiable Forth has issues due to computational complexity with larger lengths, as mentioned in the paper.

On the WAPs (**T5**), DeepProbLog reaches an accuracy between 96% and 97%, similar to Bošnjak et al. [2017] (96%).

**Probabilistic programming and deep learning** The *coin-ball* problem is a standard example in the probabilistic programming community [De Raedt and Kimmig, 2014]. It describes a game in which we have a potentially biased coin and two urns. The first urn contains a mixture of red and blue balls, and the second urn a mixture of red, blue and green balls. To play the game, we toss the coin and take a ball out of each urn. We win if both balls have the same colour, or if the coin came up heads and we have at least one red ball. We want to learn the bias of the coin (the probability of heads), and the ratio of the coloured balls in each urn. We simultaneously train one neural network to classify an image of the coin as being heads or tails (`coin/2`), and a neural network to classify the colour of the ball as being either red, blue or green (`colour/4`). These are given as RGB triples. Task **T6** is thus to learn the `game/4` predicate, requiring a combination of subsymbolic reasoning, learning and probabilistic reasoning. The input consists of an image, two RGB pairs and the output is the outcome of the game. The coin-ball problem uses a very simple neural network component. Training on a set of 256 instances converges after 5 epochs, leading to 100% accuracy on the test set (64 instances). At this point, both networks correctly classify the colours and the coins, and the probabilistic parameters reflect the distributions in the training set.

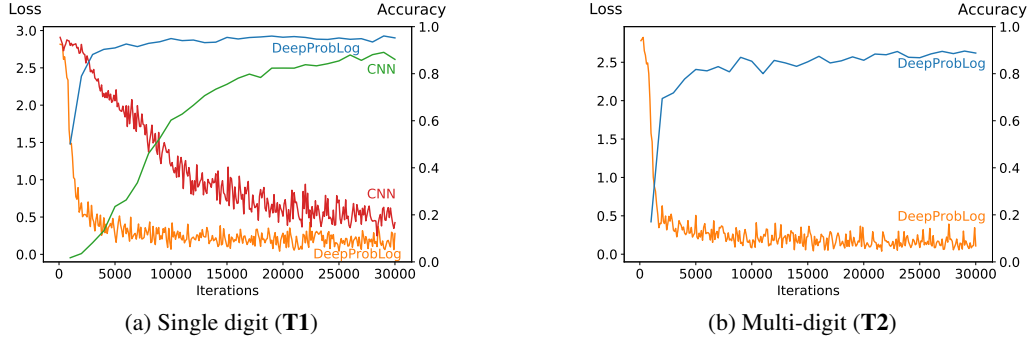

|                    |                    |
|--------------------|--------------------|
| (a) Single digit (**T1**) | (b) Multi-digit (**T2**) |

Figure 3: MNIST Addition problems: displaying training loss (red for CNN, orange for Deep-ProbLog) and F1 score on the test set (green for CNN, blue for DeepProbLog).

|  |  | Sorting (**T4**): Training length |  |  |  |  | Addition (**T3**): training length |  |  |
|---|---|---|---|---|---|---|---|---|---|
|  | Test Length | 2 | 3 | 4 | 5 | 6 | 2 | 4 | 8 |
| ∂4 [Bošnjak et al., 2017] | 8 | 100.0 | 100.0 | 49.22 | – | – | 100.0 | 100.0 | 100.0 |
|  | 64 | 100.0 | 100.0 | 20.65 | – | – | 100.0 | 100.0 | 100.0 |
| DeepProbLog | 8 | 100.0 | 100.0 | 100.0 | 100.0 | 100.0 | 100.0 | 100.0 | 100.0 |
|  | 64 | 100.0 | 100.0 | 100.0 | 100.0 | 100.0 | 100.0 | 100.0 | 100.0 |

(a) Accuracy on the addition (**T3**) and sorting (**T4**) problems (results for ∂4 reported by Bošnjak et al. [2017]).

| Training length ⟶ | 2 | 3 | 4 | 5 | 6 |
|---|---|---|---|---|---|
| ∂4 on GPU | 42 s | 160 s | – | – | – |
| ∂4 on CPU | 61 s | 390 s | – | – | – |
| DeepProbLog on CPU | 11 s | 14 s | 32 s | 114 s | 245 s |

(b) Time until 100% accurate on test length 8 for the sorting (**T4**) problem.

Table 1: Results on the Differentiable Forth experiments

**Implementation details** In all experiments we optimize the cross-entropy loss between the predicted and desired query probabilities. The network used to classify MNIST images is a basic architecture based on the PyTorch tutorial. It consists of 2 convolutional layers with kernel size 5, and respectively 6 and 16 filters, each followed by a maxpool layer of size 2, stride 2. After this come 3 fully connected layers of sizes 120, 84 and 10 (19 for the CNN baseline). It has a total of 44k parameters. The last layer is followed by a softmax layer, all others are followed by a ReLu layer. The colour network consists of a single fully connected layer of size 3. For all experiments we use Adam [Kingma and Ba, 2015] optimization for the neural networks, and SGD for the logic parameters. The learning rate is $0.001$ for the MNIST network, and $1$ for the colour network. For robustness in optimization, we use a warm-up of the learning rate of the logic parameters for the coin-ball experiments, starting at $0.0001$ and raising it linearly to $0.01$ over four epochs. For the Forth experiments, the architecture of the neural networks and other hyper-parameters are as described in Bošnjak et al. [2017]. For the Coin-Urn experiment, we generate the RGB pairs by adding Gaussian noise ($\sigma = 0.03$) to the base colours in the HSV domain. The coins are MNIST images, where we use even numbers as heads, and odd for tails. For the implementation we integrated ProbLog2 [Dries et al., 2015] with PyTorch [Paszke et al., 2017]. We do not perform actual mini-batching, but instead use gradient accumulation. All programs are listed in the appendix.

# 7 Related Work

Most of the work on combining neural networks and logical reasoning comes from the *neuro-symbolic reasoning* literature [Garcez et al., 2012, Hammer and Hitzler, 2007]. These approaches typically focus on approximating logical reasoning with neural networks by encoding logical terms in Euclidean space. However, they neither support probabilistic reasoning nor perception, and are often limited to non-recursive and acyclic logic programs [Hölldobler et al., 1999]. DeepProbLog takes a different approach and integrates neural networks into a probabilistic logic framework, retaining the full power of both logical and probabilistic reasoning and deep learning.

The most prominent recent line of related work focuses on developing differentiable frameworks for logical reasoning. Rocktäschel and Riedel (2017) introduce a differentiable framework for theorem proving. They re-implemented Prolog's theorem proving procedure in a differentiable manner and enhanced it with learning subsymbolic representation of the existing symbols, which are used to handle noise in data. Whereas Rocktäschel and Riedel (2017) use logic only to construct a neural network and focus on learning subsymbolic representations, DeepProblog focuses on tight interactions between the two and parameter learning for both the neural and the logic components. In this way, DeepProbLog retains the best of both worlds. While the approach of Rocktäschel and Riedel could in principle be applied to tasks T1 and T5, the other tasks seem to be out of scope. Cohen et al. (2018) introduce a framework to compile a tractable subset of logic programs into differentiable functions and to execute it with neural networks. It provides an alternative probabilistic logic but it has a different and less developed semantics. Furthermore, to the best of our knowledge it has not been applied to the kind of tasks tackled in the present paper. The approach most similar to ours is that of Bošnjak et al. [2017], where neural networks are used to fill in *holes* in a partially defined Forth program. DeepProblog differs in that it uses ProbLog as the host language which results in native support for both logical and probabilistic reasoning, something that has to be manually implemented in differentiable Forth. Differentiable Forth has been applied to tasks T3-5, but it is unclear whether it could be applied to the remaining ones. Finally, Evans and Grefenstette (2018) introduce a differentiable framework for rule induction, that does not focus on the integration of the two approaches like DeepProblog.

A different line of work centers around including background knowledge as a regularizer during training. Diligenti et al. [2017] and Donadello et al. [2017] use FOL to specify constraints on the output of the neural network. They use fuzzy logic to create a differentiable way of measuring how much the output of the neural networks violates these constraints. This is then added as an additional loss term that acts as a regularizer. More recent work by Xu et al. [2018] introduces a similar method that uses probabilistic logic instead of fuzzy logic, and is thus more similar to DeepProbLog. They also compile the formulas to an SDD for efficiency. However, whereas DeepProbLog can be used to specify probabilistic logic programs, these methods allow you to specify FOL constraints instead.

Dai et al. [2018] show a different way to combine perception with reasoning. Just as in DeepProbLog, they combine domain knowledge specified as purely logical Prolog rules with the output of neural networks. The main difference is that DeepProbLog deals with the uncertainty of the neural network's output with probabilistic reasoning, while Dai et al. do this by revising the hypothesis, iteratively replacing the output of the neural network with anonymous variables until a consistent hypothesis can be formed.

An idea similar in spirit to ours is that of Andreas et al. (2016), who introduce a neural network for visual question answering composed out of smaller modules responsible for individual tasks, such as object detection. Whereas the composition of modules is determined by the linguistic structure of the questions, DeepProbLog uses logic programs to connect the neural networks. These successes have inspired a number of works developing (probabilistic) logic formulations of basic deep learning primitives [Šourek et al., 2018, Dumančić and Blockeel, 2017, Kazemi and Poole, 2018].

## 8 Conclusion

We introduced DeepProbLog, a framework where neural networks and probabilistic logic programming are integrated in a way that exploits the full expressiveness and strengths of both worlds and can be trained end-to-end based on examples. This was accomplished by extending an existing probabilistic logic programming language, ProbLog, with neural predicates. Learning is performed by using aProbLog to calculate the gradient of the loss which is then used in standard gradient-descent based methods to optimize parameters in both the probabilistic logic program and the neural networks. We evaluated our framework on experiments that demonstrate its capabilities in combined symbolic and subsymbolic reasoning, program induction, and probabilistic logic programming.

## Acknowledgements

RM is a SB PhD fellow at FWO (1S61718N). SD is supported by the Research Fund KU Leuven (GOA/13/010) and Research Foundation - Flanders (G079416N)

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
