[Supplementary Material]

# A  DeepProbLog Programs

```
nn(m_digit, X, [0,...,9]) :: digit(X,0);...;digit(X,9).

addition(X,Y,Z) :- digit(X,X2), digit(Y,Y2), Z is X2+Y2.
```

Listing 1: Single-digit MNIST addition (**T1**)

In Listing 1, `digit/2` is the neural predicate that classifies an MNIST image into the integers 0 to 9. The `addition/3` predicate's first two arguments are MNIST digits, and the last is the sum. It classifies both images using and calculates the sum of the two results.

```
nn(m_digit, X, [0,...,9]) :: digit(X,0);...;digit(X,9).

number([],Result,Result).
number([H|T],Acc,Result) :-
    digit(H,Nr),
    Acc2 is Nr+10*Acc,
    number(T,Acc2,Result).
number(X,Y) :- number(X,0,Y).

multi_addition(X,Y,Z) :- number(X,X2), number(Y,Y2), Z is X2+Y2.
```

Listing 2: Multi-digit MNIST addition (**T2**)

In Listing 2, the only difference with Listing 1 is that the `multi_addition/3` predicate now uses the `number/2` predicate instead of the `digit/2` predicate. The `number/3` predicate's first argument is a list of MNIST images. It uses the `digit/2` neural predicate on each image in the list, summing and multiplying by ten to calculate the number represented by the list of images (e.g. `number([▤,▦],38)`).

```
nn(m_result,D1,D2,Carry,[0,...,9])::result(D1,D2,Carry,0);
                                    ...;result(D1,D2,Carry,9).

nn(m_carry,D1,D2,Carry,[0,1])::carry(D1,D2,Carry,0);carry(D1,D2,Carry,1).

slot(I1,I2,Carry,NewCarry,Result) :-
    result(I1,I2,Carry,Result),
    carry(I1,I2,Carry,NewCarry).

add([],[],[C],C,[]).

add([H1|T1],[H2|T2],C,Carry,[Digit|Res]) :-
    add(T1,T2,C,NewCarry,Res),
    slot(H1,H2,NewCarry,Carry,Digit).
```

Listing 3: Forth addition sketch (**T3**)

In Listing 3, there are two neural predicates: `result/4` and `carry/4`. These are used in the `slot/4` predicate that corresponds to the slot in the Forth program. The first three arguments are the two digits and the previous carry to be summed. The next two arguments are the new carry and the new resulting digit. The `add/5` predicate's arguments are: the two list of input digits, the input carry, the resulting carry and the resulting sum. It recursively calls itself to loop over both lists, calling the `slot/5` predicate on each position, using the carry from the previous step.

In Listing 4, there's a single neural predicate: `swap/3`. It's first two arguments are the numbers that are compared, the last argument is an indicator whether to swap or not. The `bubble/3` predicate performs a single step of bubble sort on its first argument using the `hole/4` predicate. The second argument is the resulting list after the bubble step, but without its last element, which is the third

```
nn(m_swap, X,[0,1]) ::swap(X,Y,0) ; swap(X,Y,1).

hole(X,Y,X,Y):-
    swap(X,Y,0).

hole(X,Y,Y,X):-
    swap(X,Y,1).

bubble([X],[],X).
bubble([H1,H2|T],[X1|T1],X):-
    hole(H1,H2,X1,X2),
    bubble([X2|T],T1,X).

bubblesort([],L,L).

bubblesort(L,L3,Sorted) :-
    bubble(L,L2,X),
    bubblesort(L2,[X|L3],Sorted).

sort(L,L2) :- bubblesort(L,[],L2).
```

Listing 4: Forth sorting sketch (**T4**)

argument. The `bubblesort/3` predicate uses the `bubble/3` predicate, and recursively calls itself on the remaining list, adding the last element on each step to the front of the sorted list.

```
permute(0,A,B,C,A,B,C).
permute(1,A,B,C,A,C,B).
permute(2,A,B,C,B,A,C).
permute(3,A,B,C,B,C,A).
permute(4,A,B,C,C,A,B).
permute(5,A,B,C,C,B,A).

swap(0,X,Y,X,Y).
swap(1,X,Y,Y,X).

operator(0,X,Y,Z) :- Z is X+Y.
operator(1,X,Y,Z) :- Z is X–Y.
operator(2,X,Y,Z) :- Z is X*Y.
operator(3,X,Y,Z) :- Y > 0, 0 =:= X mod Y,Z is X//Y.

nn(m_net1, Repr, [0,...,6])::net1(Repr,0);...;net1(Repr,6).
nn(m_net2, Repr, [0,...,3])::net2(Repr,0);...;net2(Repr,3).
nn(m_net3, Repr, [0,1])::net3(Repr,0);net3(Repr,1).
nn(m_net4, Repr, [0,...,3])::net4(Repr,0);...;net4(Repr,3).

wap(Text,X1,X2,X3,Out) :-
    net1(Text,Perm),
    permute(Perm,X1,X2,X3,N1,N2,N3),
    net2(Text,Op1),
    operator(Op1,N1,N2,Res1),
    net3(Text,Swap),
    swap(Swap,Res1,N3,X,Y),
    net4(Text,Op2),
    operator(Op2,X,Y,Out).
```

Listing 5: Forth WAP sketch (**T5**)

In Listing 5, there are four neural predicates: `net1/2` to `net4/2`. Their first argument is the input question, and the second argument are indicator variables for the choice of respectively: one of six permutations, one of 4 operations, swapping and one of 4 operations. These are implemented in the

`permute/7`, `swap/5` and `operator/4` predicates. The `wap/5` predicate then sequences these steps to calculate the result.

```
nn(m_colour,R,G,B,[red,green,blue]):: colour(R,G,B,red);
                                colour(R,G,B,green); colour(R,G,B,blue).

nn(m_coin,Coin,[heads,tails])  ::  coin(Coin,heads); coin(Coin,tails).

t(0.5):: col(1,red); t(0.5):: col(1,blue).
t(0.333):: col(2,red); t(0.333):: col(2,green); t(0.333):: col(2,blue).
t(0.5):: is_heads.

outcome(heads,red,_,win).
outcome(heads,_,red,win).
outcome(_,C,C,win).
outcome(Coin,Colour1,Colour2,loss) :- \+outcome(Coin,Colour1,Colour2,win).

game(Coin,Urn1,Urn2,Result) :-
    coin(Coin,Side),
    urn(1,Urn1,C1),
    urn(2,Urn2,C2),
    outcome(Side,C1,C2,Result).

urn(ID,Colour,C) :-
    col(ID,C),
    colour(Colour,C).

coin(Coin,heads) :-
    coin(Coin,heads),
    is_heads.

coin(Coin,tails) :-
    coin(Coin,tails),
    \+is_heads.
```

Listing 6: The coin-ball problem (**T6**)

In Listing 6, there are two neural predicates: `colour/4` and `coin/2`. There are also 6 learnable parameters: 2 for the first urn, 3 for the second and one for the coin. The `outcome/4` defines the winning conditions based on the coin and the two urns. The `urn/3` and `coin/2` predicates tie the parameters to the detections of the neural predicates. The game predicate is the high-level predicate that plays the game.