[Reviews · NeurIPS 2018]

Reviewer 1



This work extends the ProbLog language and uses the distribution of grounded facts estimated by the ProbLog to train neural networks, which is represented as neural predicates in the ProbLog. Meanwhile, the DeepProbLog framework is able to learn ProbLog parameters and deep neural networks at the same time. The experimental results show that the DeepProbLog can perform joint probabilistic logical reasoning and neural network inference on some simple tasks. Combining perception and symbolic reasoning is an important challenge for AI and machine learning. Different to most of the existing works, this work does not make one side subsumes the other (e.g. emulating logical inference with differentiable programs). Instead, the DeepProbLog handles perception and reasoning with neural nets and logic programs separately. Following are my major questions: 1. The paper says that the DeepProbLog is able to perform program induction (in experiments T3, T4, T5). However, the experiments (T1-T5) and the supplementary materials show that only neural predicates are learned from the data, while the ProbLog programs are fixed as pre-defined background knowledge; the experiment T6 learns the parameters of probabilistic facts. Is it possible for DeepProbLog to learn logic program (i.e. the logic rules)? 2. My major concern is about the scalability of DeepProbLog. Before calculating the gradients, the learning process of DeepProbLog is based on a grounding procedure, which may produce a large number of weighted logical facts. Will the scale of task affect the learning performance heavily? The experiments of combining logical reasoning and deep learning are conducted on relatively small tasks (digits of lengths ranging from 1 to 3), how would DeepProbLog perform if it is applied on domains with longer digits? Can this framework be extended to a more complex perceptual domain? 3. Finally, although this paper has covered a broad range of related works, I would like to invite the authors to comment on some other related works that may have missed by the authors. In particular, a recent paper that trains deep neural network and learns first-order logic rules jointly: Wang-Zhou Dai, Qiu-Ling Xu, Yang Yu, Zhi-Hua Zhou: Tunneling Neural Perception and Logic Reasoning through Abductive Learning. CoRR abs/1802.01173 (2018) (https://arxiv.org/abs/1802.01173) I think its motivation and techniques are very close to this work, the authors should consider comparing against or relating their method to this work. Works on Bayesian Programming by Josh Tenenbaum also learns human understandable probabilistic programs and subsymbolic perception models (although not neural nets) at the same time, which should be related to DeepProbLog as well. Minor questions: 1. The presentation of the experimental results could be improved, e.g., use task notation (T3, T4) in Table 1. Results of T5 is hidden in the text, it would be better if include them in a table. 2. For task T3-T5, are the inputs to neural predicates MNIST images? 3. The ProbLog codes in supplementary materials are confusing, please include more explanations to make it clearer. ============ After reading the rebuttal, I think the authors still have a little over claim about the contribution. "Using ProbLog to train neural predicates written in a template" is not equal to "program induction".

Reviewer 2



The paper proposes DeepProbLog, a combination of probabilistic logic programming (PLP) and neural networks. In PLP, each atom is a Boolean random variable. The integration is achieved by considering NN that output a probability distribution, such as those that have a final softmax layer. In that case, the output of a network can be associated to a PLP construct such as a probabilistic fact or annotated disjunction and, in turn, to ground facts for so called neural predicates. Inference can then be performed by computing the probability of the atoms for neural predicates by forward evaluation of the network plus knowledge compilation to sentential decision diagrams. Learning is performed by gradient descent using algebraic ProbLog and the gradient semiring. The computation of the gradient is performed by a dynamic programming algorithm that also computes the gradient of the output wrt to the neural network output, so that the gradient can be propagated to the network for its training. DeepProbLog is applied to problems combining logical and perceptual reasoning such as computing the addition of two numbers given as images, inducing programs and probabilistic programming problems. The results of the comparison with CNN for addition and differentiable Forth for program induction shows that it converges faster to a better solution. The integration of logic and NN is a hot topic nowadays. Among the many proposal, I find this one particularly interesting for its simplicity that makes it a good candidate to become a reference for future systems. The idea of associating the output of NNs to the probability of being true of ground atoms makes the integration direct and clean. Then inference and learning are obtained by propagating probabilities and gradients through the circuit and network. The integration is made possible by the use of probabilistic logic programming that attaches real values to ground atoms. DeepProbLog is correctly placed in the context of related work. In particular, Rockta¨schel and Riedel (2017) integrates logic and NN by embedding logical symbols and modifying theorem proving. Learning then consists of tuning the embeddings. DeepProbLog differs because logical reasoning is retained with its soundness guarantees. Cohen et al. (2018) is more similar to DeepProbLog but uses a language that does not have a sound probabilistic semantics. So the paper falls in a very active area but presents an approach that is nevertheless novel and promising. The experiments are particularly interesting because they show the effectiveness of DeepProbLog and because they can form benchmarks for systems aiming to integrate logical and perceptual reasoning. The paper is overall clear and well written, with a balanced presentation that is easy to follow. I have only a remark on the example in section 4, which contains some imprecisions: 1) in the text you say the program has 6 parameters, yet nodes in Figure 2c have vectors with 5 or 6 elements 2) given that 3 of the 6 parameters are fixed given the other 3, why didn't you use just 3 parameters? 3) what is the meaning of p(0.5) in Figure 2b? 4) You say: "According to the neural network, the first coin is most likely heads (p = 0.9), but is entirely unsure about the second (p = 0.5)." However, Fig 2c shows p=0.2 for side(coin2,heads) After reading the rebuttal, I think the authors answered sufficiently well the reviewers' concerns.

Reviewer 3



The authors present an approach to incorporate continuous representations through neural predicates into probabilistic logic (specifically ProbLog). Their approach is able to handle continuous input, learn the parameters of neural predicates and the probabilistic program and induce partial programs. The experimental evaluations cover all these aspects but could have more interesting intermediate results. The writing is clear but there are some key details that rely on prior work making it hard to understand this work. - #72-73 What would be an example of b1, ... bm ? Also currently it looks like there is a negative sign on b1 but I assume you intend it to be ":-". - The gradient semi-ring for aProbLog was hard to follow. I am guessing eq 4 and 5 are the identities whereas e_i is a 1-hot vector. What is the difference between L(fi) in Eqn 7 vs L(f) in Eqn 8 i.e. why do they have different forms ? - Figures 1 and 2 were immensely helpful, Could you create a corresponding figure for the equation in #147-150 ? - The example 165-178 (and corresponding fig 2) seems unnecessarily complex for an example. Couldn't the same point be communicated with just the coins ? - Figure 2c. I am not sure how to interpret the numbers at the bottom of the nodes. - Relative to prior work on aProbLog, is the key difference that the gradients are propagated through neural predicates here ? Experiments - The faster convergence and generalization of the approach really shows the power of incorporating domain-knowledge through logic programs. As you mentioned, in principle, prior work on combining logical rules with neural models could be applied here too. Would they behave any differently on this task ? For example, in this domain, would their gradient updates exactly match yours ? - It would also be interesting to see the accuracy of the digit predicate over time too. - Do you have any hypothesis on why the diff-Forth doesn't generalize to longer lengths ? Shouldn't it in principle learn the accurate swap behavior (which it did with shorter training lengths) ? - While the toy domains show the promise of this approach, I would have really liked to see a real application where such programs would be useful. The closest dataset to a real application(WAP) has comparable performance to prior work with not much discussion. Please also report the exact accuracy on this set. ================ Thanks for your response. Please provide your explanations in the final version and clarify the contribution over aProbLog.